# CALoR: Towards Comprehensive Model Inversion Defense

## Abstract

Model Inversion Attacks (MIAs) aim at recovering privacy-sensitive training data from the knowledge encoded in the released machine learning models. Recent advances in the field of MIA have significantly enhanced the attack performance under multiple scenarios, posing serious privacy risks of Deep Neural Networks (DNNs). However, the development of defense strategies against MIAs is relatively backward to resist the latest MIAs and existing defenses fail to achieve further trade-off between model utility and model robustness. In this paper, we provide an in-depth analysis from the perspective of intrinsic vulnerabilities of MIAs, comprehensively uncovering the weaknesses inherent in the basic pipeline, which are partially investigated in the previous defenses. Building upon these new insights, we propose a robust defense mechanism, integrating *Confidence Adaptation* and *Low-Rank compression* (**CALoR**). Our method includes a novel robustness-enhanced classification loss specially-designed for model inversion defenses and reveals the extraordinary effectiveness of compressing the classification header. With CALoR, we can mislead the optimization objective, reduce the leaked information and impede the backpropagation of MIAs, thus mitigating the risk of privacy leakage. Extensive experimental results demonstrate that our method achieves state-of-the-art (SOTA) defense performance against MIAs and exhibits superior generalization to existing defenses across various scenarios.

## 1 Introduction

In recent years, Deep Neural Networks (DNNs) have achieved remarkable advancements, leading to significant success in a wide range of applications, such as face recognition (He et al., 2016), audio recognition (Conneau et al., 2020), and brain decoding (Chen et al., 2024). However, the powerful capabilities that make these models so effective also render them vulnerable to privacy attacks. One of the most critical threats to privacy and security is the Model Inversion Attack (MIA) (Fredrikson et al., 2015), which allows adversaries to reconstruct privacy-sensitive features from the output information of released models. For example, in face recognition systems, MIAs can produce synthetic images that reveal specific visual characteristics of the private identities even without direct access to the private training dataset. This poses a significant risk, as MIAs may allow unauthorized individuals reconstruct valid facial features to disguise as authorized personnel, thereby compromising both privacy and security.

As recent MIAs have experienced astonishing evolution in attack capabilities, however, existing defense strategies are backward to resist advanced attacks (Yuan et al., 2023; Qiu et al., 2024). Most defenses fail to enhance robustness while maintaining high utility for target models when defending the latest MIAs. Moreover, to the best of our knowledge, no comprehensive analysis has been conducted on the inherent weaknesses of MIAs in the previous defense researches. To address this gap, we dive into the nature of MIAs and present the first comprehensive and in-depth analysis of their intrinsic vulnerabilities. We concentrate on the key aspects in the basic pipeline of MIAs, including *attack objective*, *MI overfitting*, and *optimization*. By jointly interrupting these weaknesses, the defender can comprehensively improve the model robustness when encountering multiple different types of MIAs. Building on these new insights, we propose a new defense framework that exploits all the inherent weaknesses, including a **Confidence Adaptation** and a **Low-Rank compression** strategies (**CALoR**). Specifically, we design a novel confidence adaptation loss that slightly reduces the confidence of private samples, introducing bias between the attacker's optimization goal and the

valid attack objective. To prevent attackers from mitigating the MI overfitting problem which may cause a fail attack, we propose the low-rank compression strategy to compresses high-dimensional features into low dimensions, reducing the leaked information from model outputs. Additionally, we incorporate a $\mathrm{Tanh}$ activation function to induce gradient vanishing, which is utilized to impeding the attack optimization. We conduct comprehensive experiments across multiple settings to evaluate our method. The results demonstrate that our approach achieves state-of-the-art (SOTA) defense performance. In scenarios where target models demonstrate strong performance, existing defense methods often struggle against advanced attacks, while our method maintains superior defense capabilities. Extensive experiments and ablation studies thoroughly validate the effectiveness of our proposed method. In summary, our main contributions are as follows:

- We are the first to conduct comprehensive analyses of weaknesses inherent in MIAs, considering the following critical aspects: *attack objective*, *MI overfitting*, and *optimization*.
- We propose a comprehensive defense strategy, integrating ***Confidence Adaptation*** and ***Low-Rank compression*** (**CALoR**), which defends MIAs by adding bias on attack objective, reducing leaked information in model outputs and enhancing the gradient vanishing problem.
- We conduct extensive experiments to demonstrate that our proposed CALoR achieves SOTA robustness against MIAs under various scenarios, particularly under challenging high-resolution settings where the target models maintain strong performance.

## 2 BACKGROUND AND RELATED WORKS

### 2.1 MODEL INVERSION ATTACKS

Let $f_\theta : \mathcal{X} \to [0, 1]^N$ denote a general classifier, which processes a private image $\mathbf{x} \sim \mathbb{P}(X)$ and computes a prediction $\hat{y}_c \in [0, 1]$ for each class $c \in \{0, 1, \dots, N\}$. MIAs aim to reconstruct images that reveal private characteristic features of a specific identity $c$. In white-box scenarios, attackers have full access to the classifier's weights and outputs, allowing them to compute gradients across the classifier when executing the attack.

In recent years, Generative Adversarial Network (GAN) based model inversion attacks, first proposed by GMI (Zhang et al., 2020), have become the standard paradigm for MIAs (Fang et al., 2024). Specifically, attackers initially train a generator $G$ to capture a similar structural prior as the target private data. In the attack stage, they attempt to reconstruct private images $\mathbf{x}^* = G(\mathbf{z}^*)$ labeled with class $c$ to approximate the private distribution $\mathbb{P}(X)$, by optimizing the latent vectors $\mathbf{z}$:

$$\mathbf{z}^* = \arg \min_{\mathbf{z}} \mathcal{L}_{cls}(f_\theta(G(\mathbf{z})), c) + \lambda \mathcal{L}_{prior}(\mathbf{z}; G), \tag{1}$$

where $\mathcal{L}_{cls}$ is the classification loss, $\lambda$ is a hyperparameter, and $\mathcal{L}_{prior}$ denotes the optional prior knowledge regularization terms, such as the discriminator loss (Zhang et al., 2020; Chen et al., 2021) and the feature regularization loss (Nguyen et al., 2023).

Subsequent MIA studies have largely followed this pipeline and improved upon it. Researchers like Chen et al. (2021); Yuan et al. (2023) train target-specific GANs to capture more private information from the target classifier. An et al. (2022); Struppek et al. (2022); Qiu et al. (2024) utilize well-structured StyleGANs (Karras et al., 2020) to generate high-quality and high-resolution samples. Additionally, Struppek et al. (2022); Nguyen et al. (2023); Yuan et al. (2023) have explored various classification losses to mitigate the effects of gradient vanishing issue.

### 2.2 MODEL INVERSION DEFENSES

Model inversion defenses aim at reducing the threat of MIAs. Wang et al. (2021b) propose the first MI-specific defense method MID, defending attacks by reducing the mutual information between model input and output. Peng et al. (2022) upgrades from MID with a bilateral dependency optimization (BiDO). They minimize the dependency between inputs and the intermediate representations while maximizing that between the intermediate representations and outputs. Struppek et al. (2024) explores the connection between label smoothing and model robustness. They indicate that the label smoothing trick with negative factor can enhance the robustness. Ho et al. (2024) first pre-trains the model on public datasets. Then, they freeze the parameters in previous layers when fine-tuning on private datasets. More details about the defenses are described in Appendix C.

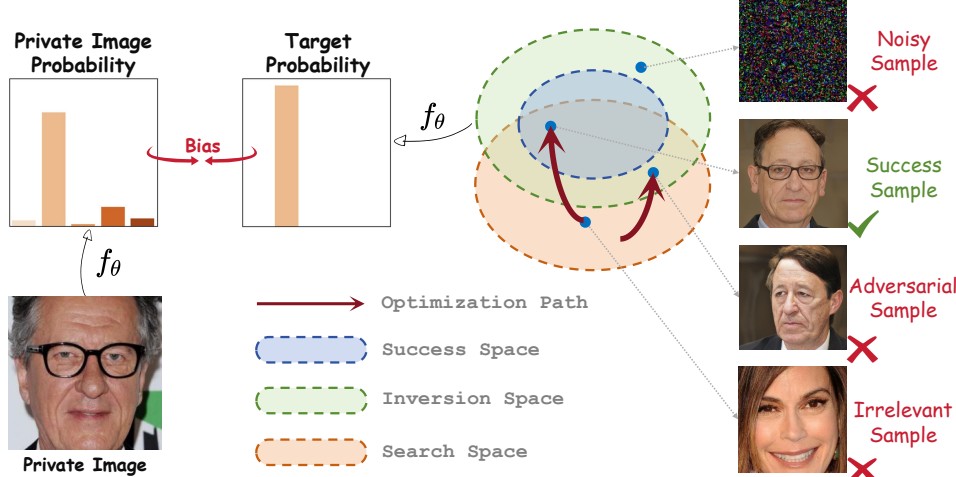

Figure 1: Overview of the inherent weaknesses of model inversion attacks. (1) Attack objective: The probability predicted by the classifier $f_\theta$ is not exactly equal to the one-hot vector, which is the optimization objective of MIAs. (2) MI overfitting: The input images that correspond to approximately one-hot predictions form the inversion space. Only a part of them can be viewed as the success samples to leak privacy information, denoted as the success space. The attacks may overfit to the target model, resulting in adversarial samples, which fail to extract privacy-sensitive information. (3) Optimization: In white-box scenarios, the attackers search images with the gradient in their search space, trying to optimize to the success samples. However, gradient optimization on DNNs suffers from gradient vanishing issue.

## 3 METHODOLOGY

### 3.1 RE-THINKING THE INHERENT WEAKNESSES OF MODEL INVERSION ATTACKS

In this section, we will illustrate the inherent weaknesses of MIAs, with an overview in Fig. 1.

**Attack Objective.** Given a target classifier $f_\theta$ and a private image $\mathbf{x}_{\text{pri},c}$ labeled with class $c$, the forward classification process is defined as $\mathbf{p}_{\text{pri},c} = f_\theta(\mathbf{x}_{\text{pri},c})$, where $\mathbf{p}_{\text{pri},c}$ represents the classifier's prediction probability vector for the private image. In contrast, attackers attempt to perform an inverse process, denoted as $\mathbf{x}_{\text{inv},c} = f_\theta^{-1}(\mathbf{p}_c)$. Here, $\mathbf{p}_c$ is the target one-hot probability vector representing the target class $c$, and $\mathbf{x}_{\text{inv},c}$ is the reconstructed image. The attackers' goal is to minimize the distinction between the private image $\mathbf{x}_{\text{pri},c}$ and the reconstructed image $\mathbf{x}_{\text{inv},c}$. However, the $\mathbf{p}_{\text{pri},c}$ is not exactly identical to the one-hot $\mathbf{p}_c$, whose confidence on other identities remains non-zero. In this case, as the prediction of the reconstructed image approaches $\mathbf{p}_c$, the recovered image may deviate further from the original private image. However, to the best of our knowledge, no existing defense mechanisms have effectively exploited such vulnerability.

**MI Overfitting.** The attacker's optimization goal is to manipulate the prediction to resemble the target one-hot probability vector. However, as illustrated in Fig. 1, despite massive inputs can meet the optimization goal, only a subset of them can accurately capture privacy-sensitive characteristics, which are defined as success samples. The other inputs fail to reveal private information due to the MI overfitting problem (Nguyen et al., 2023). It means that the reconstructed samples become overly tailored to random variations and noise in the target model's parameters. Consequently, the MI overfitting problem results in a failure to correctly uncover the private characteristics of the target identity. While attackers can reduce MI overfitting by excluding noisy samples that are not conform to the basic structure of private data with a generative prior, they cannot thoroughly avoid generating adversarial samples. These adversarial samples tend to overfit to the specific patterns in the target model, failing to expose the correct private information and thus leading to unsuccessful attacks. Nevertheless, the MI overfitting phenomenon may be beneficial for defenses once utilized properly.

**Optimization.** In white-box scenarios, the attacker employs gradient-based methods for optimization to extract the information encoded in the model parameters. However, performing optimization

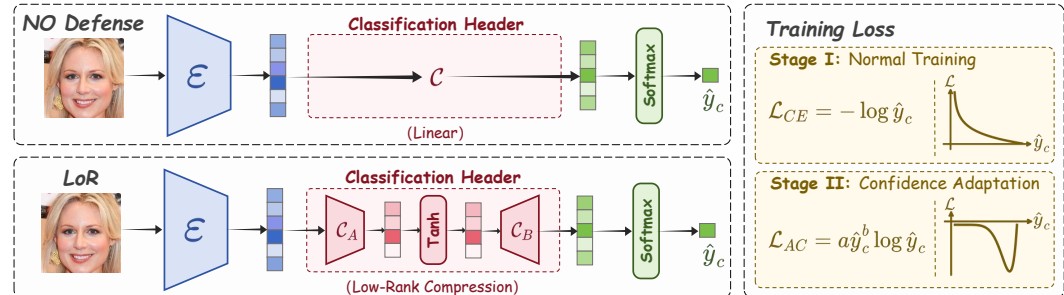

Figure 2: Overview of our CALoR defense strategy. Unlike a standard classifier, we replace the classification header with a low-rank compression (LoR) header. The features extracted by the encoder $\mathcal{E}$ are compressed into a lower-dimensional space by a linear layer $\mathcal{C}_A$, followed by a $\mathrm{Tanh}$ activation function. The compressed features are then passed through another linear layer $\mathcal{C}_B$ for classification prediction. The model is initially trained using cross-entropy loss ($\mathcal{L}_{CE}$), followed by fine-tuning with an confidence adaptation loss ($\mathcal{L}_{CA}$).

.

in DNNs faces several challenges (Struppek et al., 2022). The non-convex nature of the optimization landscape (Dauphin et al., 2014) often leads to the optimization process becoming trapped in suboptimal local minima. Additionally, the gradient vanishing problem (Glorot et al., 2011) significantly hampers the optimization by increasing difficulty in efficient backpropagation, exhibiting the potential application in the defenses.

Previous approaches typically utilize only one single vulnerability and lack a comprehensive defense strategy, as discussed in Appendix C. To address the aforementioned limitations, we propose a robust defense mechanism that integrates confidence adaptation (Sec. 3.2) and a low-rank compression strategy (Sec. 3.3). An overview of our defense is provided in Fig. 2.

## 3.2 CONFIDENCE ADAPTATION

As outlined in Section 2.1, minimizing the classification loss is a primary optimization objective in MIAs, which means pushing the target classifier to output an approximately one-hot probability vector $\mathbf{p}_c$, i.e., $f_\theta(\mathbf{x}_{\text{inv}}) = \mathbf{p}_c$. This exploits an assumption that the probability vector for the private image $\mathbf{p}_{\text{pri}}$ and the target probability vector $\mathbf{p}_c$ are highly similar. Therefore, a straightforward defense against MIAs is to add appropriate bias between $\mathbf{p}_{\text{pri}}$ and $\mathbf{p}_c$. Specifically, we can deploy the defense by reducing the prediction confidence, which is defined as the predicted probability for the label $c$. To validate this analysis, we train a series of classifiers with varying levels of average classification confidence on private images. The details are provided in Appendix I.1. Then, we apply PPA (Struppek et al., 2022) and PLG (Yuan et al., 2023) attacks to these models, evaluating the attack accuracy for both attacks. The

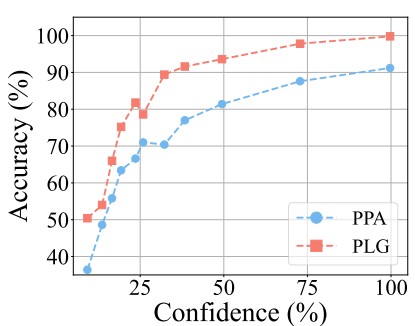

Figure 3: The attack accuracy of PPA and PLG on models with different average confidence on private images.

results are presented in Fig. 3, which demonstrate that lower confidence brings more challenging to MIAs. Motivated by this finding, we propose a two-stage training strategy to introduce the bias without significantly degrading the model utility. In the first stage, we train the model with the standard cross-entropy loss to obtain adequate model performance. During the second stage, we design a novel confidence adaptation loss to fine-tune the model for enhancing more robustness. The loss function is as follows:

$$\mathcal{L}_{CA} = a\hat{y}_c^b \log \hat{y}_c, \tag{2}$$

where $a > 0$ and $b > 0$ are hyperparameters, and $\hat{y}_c$ represents the predicted probability of the target class $c$, referred to as confidence. Upon full convergence of the optimization, $\hat{y}_c$ will converge to

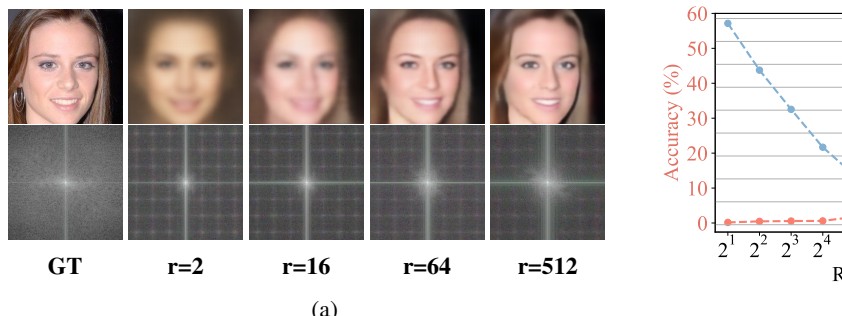

GT  r=2  r=16  r=64  r=512

(a)            (b)

Figure 4: Experiment results of autoencoders with varying ranks $r$. (a) Comparison of ground truth (GT) and reconstructed images in the spatial and frequency domains. As $r$ decreases, the reconstruction is concentrated in the low-frequency regions, capturing only the basic spatial structure with minimal detail restoration. (b) Evaluation of reconstructed images. It contains mean square error (MSE) loss between input and reconstructed images, and the re-classification accuracy.

$\exp(-1/b) < 1$ [1]. By introducing this loss, we slightly reduce the confidence of private samples in the model. This moves them away from the attack optimization target of confidence value 1, thus effectively protecting private samples.

### 3.3 LOW-RANK COMPRESSION

As discussed in Section 3.1, some samples fail to reveal critical privacy attributes owing to the MI overffiting issue to the target models. To achieve effective attack outcomes, recent attacks attempt to extract more information from the target classifier's outputs (*e.g.*, by employing data augmentation and querying the target model multiple times (Struppek et al., 2022)), thereby increasing the likelihood that the attack results are success samples. To mitigate this, an effective defense mechanism is to limit the amount of input-related information encoded in the model's output. Previous works like MID (Wang et al., 2021b) and BiDO (Peng et al., 2022), have explored this approach, though they often face significant performance trade-offs to achieve sufficient defense capabilities.

Inspired by the feature compression employed in autoencoders , we initiate our study by examining the impact of compressed ranks on the preservation of input information. An autoencoder consists of two components: an encoder and a decoder. The encoder compresses the input image $\mathbf{x}$ into a latent vector $\mathbf{z} \in \mathbb{R}^r$, while the decoder attempts to reconstruct the image from $\mathbf{z}$. We train a series of autoencoders on the CelebA dataset (Liu et al., 2015) with varying latent dimensions (rank $r$) and evaluate them on the test split. The details of the autoencoder experiments are shown in Appendix I.2. We compute the reconstruction mean square error (MSE) loss and re-classification accuracy of the reconstructed samples. The evaluation results are shown in Fig. 4b, with visual samples displayed in Fig. 4a. The results show that lower compressed ranks lead to higher reconstruction error and reduced image utility, indicating greater information loss from the input image. This suggests that a compression into lower ranks reduces more input information. This information reducing approach is crucial for designing models robust against MIAs (Peng et al., 2022).

Motivated by aforementioned findings, we propose a low-rank compression strategy to address this challenge. It encourages the model to retain only the information essential for classification while discarding irrelevant details. In practice, we decompose the classifier into two components: $f_\theta = \mathcal{C} \circ \mathcal{E}$, where $\mathcal{E}$ is the encoder, and $\mathcal{C}$ is the classification header. In standard classifiers, $\mathcal{C}$ is a linear layer with a weight matrix $\mathbf{W}^{m \times N}$, where $m$ denotes the dimensionality of $\mathcal{C}$'s input features and $N$ is the number of classes. To reduce the information contained in the model outputs, We decompose $\mathcal{C}$ into two linear layers, $\mathcal{C}_A$ and $\mathcal{C}_B$. The weight matrices for these layers are $\mathbf{W}_A^{m \times r}$ and $\mathbf{W}_B^{r \times N}$, respectively. Here, $r \ll \min\{m, N\}$. This decomposition satisfies $\mathbf{W}^{m \times N} = \mathbf{W}_A^{m \times r} \mathbf{W}_B^{r \times N}$. In simpler terms, $\mathcal{C}_A$ compresses the $m$-dimensional feature space into $r$ dimensions, and $\mathcal{C}_B$ then uses this compressed representation to make the final classification predictions.

---

[1]The proof of this convergence value is provided in Appendix B.

Moreover, Glorot et al. (2011) emphasize that certain non-linear activation functions, such as Sigmoid and Tanh, are susceptible to the gradient vanishing problem. To further hinder the attacker's optimization process, we propose introducing a non-linear activation function after the compressed features, specifically positioned between the two linear layers of the classification head. The back-propagation characteristics of various activation functions suggest that the Tanh function is particularly prone to causing the gradient vanishing problem, which will be analyzed in detail in Appendix J. As a result, we choose Tanh as the activation function in our approach.

Following Struppek et al. (2024), we also provide a toy example of a simple MIA in Appendix A. It offers a more intuitive illustration of the role of low-rank compression against MIAs.

## 4 EXPERIMENT

### 4.1 EXPERIMENTAL PROTOCOL

**Datasets.** Following previous works, we focus on the facial recognition task on private datasets $\mathcal{D}_{\text{pri}}$, including FaceScrub (Ng & Winkler, 2014) and CelebA (Liu et al., 2015). The two datasets contains images of $530$ and $10177$ different identities, respectively. For the CelebA dataset, only top-1000 identities with the most samples will be used (Zhang et al., 2020). The images are pre-processed to $64 \times 64$ and $224 \times 224$ in low- and high-resolution scenarios, respectively.

**Models.** We trained classifiers for multiple architectures, including convolution-based IR-152 (He et al., 2016) , ResNet-152 (He et al., 2016) and FaceNet-112 (Cheng et al., 2017), as well as transformer-based ViT-B/16 (Dosovitskiy et al., 2020), Swin-v2 (Liu et al., 2022) and MaxViT (Tu et al., 2022). FaceNet-112 and MaxViT are used as evaluation models for the low- and high-resolution cases, respectively. The remaining models are trained under different defense algorithms and used as target models of different attacks. Target models are pre-trained with a public dataset $\mathcal{D}_{\text{pre}}$, such as MS-Celeb-1M (Guo et al., 2016) and ImageNet (Deng et al., 2009). Then they will be fine-tuned on the private dataset. More details of model training are provided in Appendix D.1.

**Attacks.** We compare various MI attack methods in our experiments, including GMI (Zhang et al., 2020), KED (Chen et al., 2021), Mirror (An et al., 2022), PPA (Struppek et al., 2022), LOMMA (Nguyen et al., 2023), PLG (Yuan et al., 2023) and IF (Qiu et al., 2024). More details of attacks are shown in Appendix D.2.

**Metrics.** Following the previous works (Struppek et al., 2022), we employ various metrics to evaluate the effectiveness of different defense methods, mainly including attack accuracy and feature distance. In addition, we also analyze some other aspects of model robustness in Appendix F.

*Attack Accuracy.* We use the evaluation model to predict the labels on reconstructed samples and compute the top-1 and top-5 accuracy for target classes, denoted as **Acc@1** and **Acc@5** respectively. Higher attack accuracy indicates a greater leakage of private information (Zhang et al., 2020).

*Feature Distance.* Features are defined as the output from the model's penultimate layer. For each reconstructed sample, we calculate the $l_2$ feature distance to the nearest private training sample. The final metric is obtained by averaging these distances across all reconstructed samples. The feature distances are evaluated using the evaluation model and a FaceNet model (Schroff et al., 2015) trained on a large VGGFace2 dataset (Cao et al., 2018), denoted as $\delta_{eval}$ and $\delta_{face}$, respectively.

### 4.2 COMPARISON WITH PREVIOUS DEFENSE METHODS

We compare our method with $4$ SOTA defense methods, including MID (Wang et al., 2021b), BiDO (Peng et al., 2022), LS (Struppek et al., 2024) and TL (Ho et al., 2024). To ensure a fair comparison among the different defense algorithms, we carefully adjust the hyperparameters governing the defense strength, maintaining nearly identical classification accuracy on the test set (**Test Acc**) for each model. Detailed implementation of the defense methods is provided in Appendix D.1. In this section, we present the results of two most advanced attack methods: IF, which is based on an unconditional GAN, and PLG, which employs a conditional GAN.

In the low-resolution scenario, we provide quantitative attack results against IR-152 trained on the FaceScrub dataset in Table 1. We can observe that our method achieves significant improvements

Table 1: IF and PLG attack results against IR-152 models trained on FaceScrub dataset in the low-resolution scenario. Following previous studies (Zhang et al., 2020; Yuan et al., 2023), the models are pre-trained with MS-Celeb-1M and fine-tuned on FaceScrub dataset.

| Method | Test Acc | IF | | | | PLG | | | |
|---|---|---|---|---|---|---|---|---|---|
| | | ↓**Acc@1** | ↓**Acc@5** | ↑$\delta_{eval}$ | ↑$\delta_{face}$ | ↓**Acc@1** | ↓**Acc@5** | ↑$\delta_{eval}$ | ↑$\delta_{face}$ |
| **NO Defense** | 98.2% | 82.2% | 88.4% | 1376.96 | 0.70 | 100.0% | 100.0% | 907.75 | 0.47 |
| **MID** | 97.0% | 78.8% | 87.0% | 1479.14 | 0.75 | 99.2% | 99.6% | 1062.76 | 0.54 |
| **BiDO** | 95.2% | 51.0% | 71.0% | 1748.32 | 0.81 | 82.2% | 92.6% | 1533.35 | 0.63 |
| **LS** | 97.3% | 81.8% | 87.6% | 1391.75 | 0.74 | 99.8% | 100.0% | 839.54 | 0.46 |
| **TL** | 95.4% | 42.0% | 55.6% | 1903.27 | 0.94 | 93.6% | 97.8% | 1202.20 | 0.54 |
| **CALoR(ours)** | 97.0% | **10.0%** | **23.6%** | **2337.56** | **1.29** | **48.2%** | **63.0%** | **1999.24** | **0.98** |

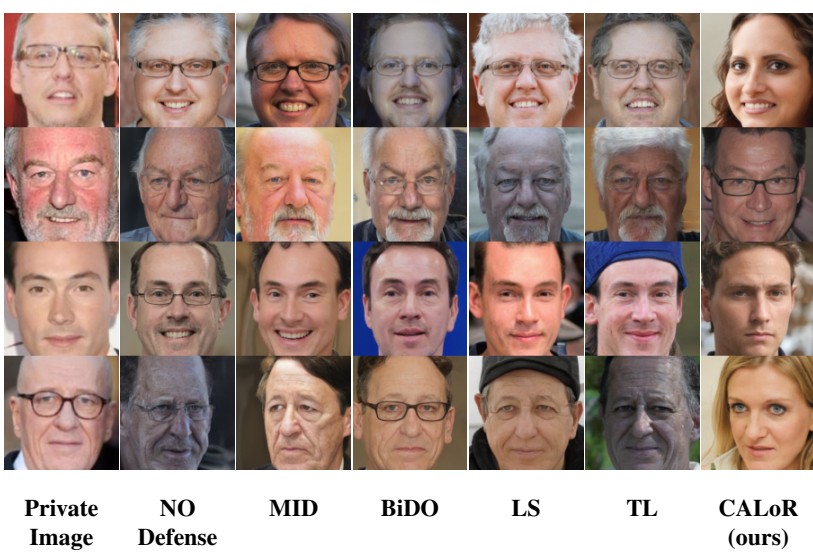

| **Private Image** | **NO Defense** | **MID** | **BiDO** | **LS** | **TL** | **CALoR (ours)** |

Figure 5: Visual comparison of IF attacks against ResNet-152 under different defense strategies.

over previous defense methods. Especially in terms of attack accuracy, our method can reduce it by more than 30% compared with the best previous defense algorithm. We also conduct additional experiments with other attack methods. The results presented in Appendix E show that our approach demonstrates effective defense across a range of attacks.

In high-resolution scenarios, previous studies (Struppek et al., 2022; 2024; Qiu et al., 2024) have utilized target models that are pre-trained on ImageNet and subsequently fine-tuned on the private dataset. However, this approach results in models with insufficient accuracy, which does not accurately represent real-world conditions (Sevastopolskiy et al., 2023). To bridge this gap, we pre-train the models on a large face dataset MS-Celeb-1M, as in the low-resolution scenario. This pre-training process allows the models to achieve a test accuracy of over 96% on the FaceScrub dataset. We conduct experiments under two conditions using different pre-training datasets, with the attack results presented in Table 2. Our method demonstrates superior defense performance. When models show low accuracy with ImageNet backbone, consistent with previous studies, recent defense methods such as LS and TL demonstrate improved defense capabilities. However, our approach still achieves the best defensive performance. Notably, in scenarios with high classification accuracy with MS-Celeb-1M backbone, previous methods show significant difficulty in defending against the most advanced attacks. In contrast, our method reduces the attack accuracy of IF and PLG by 38.4% and 52.0%, respectively, and increases the feature distance between the reconstructed and private images to 1.5 to 2 times compared to the case without defense. The visualization results of the reconstructed images from IF attacks under various defense methods are shown in Fig. 5. Compared to previous approaches, our defense strategy significantly increases the disparity between the reconstructed images and the original private images.

Table 2: IF and PLG attack results against ResNet-152 models trained on FaceScrub dataset in the high-resolution scenario. $\mathcal{D}_{pre}$ means the public dataset to pre-train the model, including ImageNet (IN) and MS-Celeb-1M (MS) .

| $\mathcal{D}_{pre}$ | Method | Test Acc | IF | | | | PLG | | | |
|---|---|---|---|---|---|---|---|---|---|---|
| | | | $\downarrow$ **Acc@1** | $\downarrow$ **Acc@5** | $\uparrow \delta_{eval}$ | $\uparrow \delta_{face}$ | $\downarrow$ **Acc@1** | $\downarrow$ **Acc@5** | $\uparrow \delta_{eval}$ | $\uparrow \delta_{face}$ |
| IN | **NO Defense** | 92.2% | 85.8% | 96.8% | 376.76 | 0.71 | 81.4% | 97.6% | 405.25 | 0.74 |
| | **MID** | 88.2% | 81.2% | 90.0% | 392.07 | 0.77 | 55.8% | 74.6% | 468.09 | 0.92 |
| | **BiDO** | 88.6% | 79.2% | 92.4% | 391.86 | 0.73 | 54.8% | 78.8% | 464.53 | 0.85 |
| | **LS** | 88.8% | 29.6% | 57.4% | 508.42 | 1.01 | 23.8% | 43.8% | 559.46 | 1.13 |
| | **TL** | 88.4% | 64.6% | 85.0% | 420.23 | 0.81 | 12.6% | 32.2% | 586.17 | 1.19 |
| | **CALoR(ours)** | 89.4% | **28.2%** | **55.6%** | **512.05** | **1.04** | **11.4%** | **22.2%** | **623.32** | **1.32** |
| MS | **NO Defense** | 98.5% | 99.2% | 99.6% | 285.59 | 0.52 | 99.2% | 100.0% | 311.98 | 0.55 |
| | **MID** | 96.8% | 99.8% | 100.0% | 250.76 | 0.46 | 99.6% | 100.0% | 287.33 | 0.53 |
| | **BiDO** | 96.3% | 98.4% | 99.8% | 302.38 | 0.55 | 98.4% | 99.8% | 339.76 | 0.58 |
| | **LS** | 96.4% | 99.0% | 100.0% | 295.42 | 0.55 | 64.6% | 73.2% | 447.83 | 0.92 |
| | **TL** | 96.6% | 100.0% | 100.0% | 235.14 | 0.42 | 98.6% | 99.8% | 335.53 | 0.59 |
| | **CALoR(ours)** | 96.8% | **60.8%** | **77.4%** | **445.44** | **0.89** | **47.2%** | **68.8%** | **497.83** | **1.00** |

In addition to the previously discussed experiments, we conduct further comparative studies under a wider range of experimental settings. These additional settings include more attack methods, more private datasets such as CelebA, and more target classifier architectures, including transformer-based models like Swin-v2 and ViT-B/16. The experimental results shown in Appendix E demonstrate that our defense method consistently achieves strong performance across a variety of settings.

## 4.3 ABLATION STUDY

In this section, we conduct experiments to explore how effectively our method defends against the inherent weaknesses of MIAs.

**Confidence adaptation.** To quantitatively evaluate our confidence adaptation strategy, we propose an ablation experiment on IF attacks against IR-152 and ResNet-152 in both low- and high-resolution scenarios. The results listed in Table 3 indicate that confidence adaptation loss can reduce the attack performance without significantly affecting the utility of the models. Especially, the loss function reduces the accuracy of IF attacks by 3% and 4%.

Table 3: The influence of confidence adaptation loss on IF attacks. LoR implies that the confidence adaptation loss is not used.

| Method | Low Resolution | | | | | High Resolution | | | | |
|---|---|---|---|---|---|---|---|---|---|---|
| | **Test Acc** | $\downarrow$ **Acc@1** | $\downarrow$ **Acc@5** | $\uparrow \delta_{eval}$ | $\uparrow \delta_{face}$ | **Test Acc** | $\downarrow$ **Acc@1** | $\downarrow$ **Acc@5** | $\uparrow \delta_{eval}$ | $\uparrow \delta_{face}$ |
| **NO Defense** | 98.2% | 82.2% | 88.4% | 1376.96 | 0.70 | 98.5% | 99.2% | 99.6% | 285.59 | 0.52 |
| **LoR** | **97.1%** | 13.0% | 27.0% | 2299.39 | **1.29** | 96.6% | 64.8% | 83.2% | 431.29 | 0.86 |
| **CALoR** | 97.0% | **10.0%** | **23.6%** | **2337.56** | **1.29** | **96.8%** | **60.8%** | **77.4%** | **445.44** | **0.89** |

**The rank of the classifier head.** To investigate the effect of the classification header's rank on MIAs, we train a series of IR-152 and ResNet-152 models with varying ranks and evaluate their test accuracy. Subsequently, we apply the IF attacks on these models and record the attack accuracy and distance metric results. Note that to eliminate any potential interference from our confidence adaptation loss, we excluded it from these experiments. The results in low-resolution scenarios are provided in Table 4. For high-resolution scenarios, the results for ResNet-152 models are available in Appendix G. The findings reveal that a low rank, such as 30, is sufficient for the model to effectively capture essential features necessary for the classification task, leading to strong overall performance. In this context, the model exhibits notable robustness against attacks, achieving a low attack accuracy of only 13.0%. Moreover, While increasing the rank results in only marginal gains in model performance, it substantially enhances the attacker's effectiveness, raising significant privacy leakage concerns. These observations indicate that adopting a low-rank compression strategy not only maintains high model performance, but also serves as an effective defense against model inversion attacks. This dual benefit highlights the potential of low-rank approaches in ensuring both efficacy and security in model deployment.

Table 4: IF attack results against IR-152 models with different ranks in low-resolution scenarios.

| Rank | Test Acc | $\downarrow$ Acc@1 | $\downarrow$ Acc@5 | $\uparrow \delta_{eval}$ | $\uparrow \delta_{face}$ |
|------|----------|---------|---------|-------------|-------------|
| 10   | 76.2%    | 1.4%    | 5.4%    | 2653.01     | 1.49        |
| 15   | 92.3%    | 5.0%    | 9.6%    | 2527.82     | 1.41        |
| 20   | 96.1%    | 4.4%    | 11.8%   | 2475.01     | 1.40        |
| 30   | 97.1%    | 13.0%   | 27.0%   | 2299.39     | 1.29        |
| 50   | 97.3%    | 15.4%   | 30.8%   | 2218.87     | 1.18        |
| 100  | 98.5%    | 48.2%   | 66.4%   | 1805.82     | 0.97        |
| 256  | 98.4%    | 67.6%   | 79.8%   | 1669.08     | 0.84        |
| 512  | 98.6%    | 77.8%   | 86.8%   | 1518.76     | 0.79        |

**Gradient vanishing of the non-linear activation.** We begin by analyse the gradient vanishing effect of different non-linear activation functions. We apply IF attacks on ResNet-152 models in the high-resolution scenario, using different non-linear activation functions. During this process, we record the gradient magnitudes at each optimization step. The gradients values are smoothed using a window size of 20 and normalized by dividing by the first value, as displayed in Fig. 6. The results indicate a rapid decline in gradient magnitude when non-linear activation functions are applied to the model, with the $\mathrm{Tanh}$ exhibiting the most significant decrease. Table 5 presents the IF attack outcomes for each activation function. The results also reveal that the $\mathrm{Tanh}$ activation achieves an optimal balance between model performance and model inversion defense effectiveness.

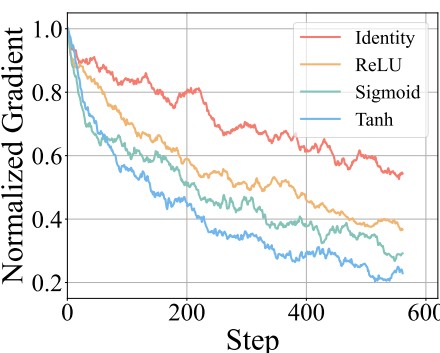

Figure 6: The trend of gradient magnitude in IF attacks. Identity refers to the absence of the activation function.

Table 5: The influence of different non-linear activation functions on IF attacks.

| Activation | Low Resolution | | | | | High Resolution | | | | |
|------------|----------|---------|---------|-------------|-------------|----------|---------|---------|-------------|-------------|
|            | Test Acc | $\downarrow$ Acc@1 | $\downarrow$ Acc@5 | $\uparrow \delta_{eval}$ | $\uparrow \delta_{face}$ | Test Acc | $\downarrow$ Acc@1 | $\downarrow$ Acc@5 | $\uparrow \delta_{eval}$ | $\uparrow \delta_{face}$ |
| **Identity** | 94.1%  | 12.6%   | 24.2%   | 2304.35     | 1.30        | 96.7%    | 82.0%   | 91.6%   | 412.29      | 0.78        |
| **ReLU**     | 92.7%  | 14.2%   | 21.8%   | 2388.95     | 1.29        | 96.1%    | 81.6%   | 93.2%   | 401.77      | 0.79        |
| **Sigmoid**  | 93.2%  | 7.4%    | 16.2%   | 2405.07     | 1.39        | **96.8%** | 67.6%  | 84.8%   | 417.11      | 0.83        |
| **Tanh**     | **96.1%** | **4.4%** | **11.8%** | **2475.01** | **1.40** | 96.6%   | **64.8%** | **83.2%** | **431.29** | **0.86**    |

## 5 CONCLUSION

Deep Neural Networks (DNNs) has tremendous potential across various domains. However, its applications must ensure robust privacy protections to prevent user privacy leakage, which can be compromised by model inversion attacks (MIAs). In this paper, we conduct a comprehensive analysis of the vulnerabilities inherent to MIAs by examining three key dimensions: *attack objective*, *MI overfitting*, and *optimization*. To address the risks posed by MIAs, we introduce a novel defense framework that combines **C**onfidence **A**daptation with a **Lo**w-**R**ank compression strategy (**CALoR**), providing a robust defense mechanism. Our CALoR comprehensively defend MIA by introducing bias between the attack objective and the goal of attack optimization, reducing information encoded in the outputs of the model to strengthen the MI overfitting problem, and letting the attacker fall into the gradient vanishing problem. Through extensive experimental validation, CALoR demonstrates SOTA effectiveness across diverse scenarios. Notably, it excels at protecting models while maintaining high performance, surpassing prior defense techniques. We hope this work provides valuable insights for mitigating privacy leakage risks in deployed models and fosters a deeper understanding of defense mechanisms against model inversion attacks.

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

# Appendix Table of Contents

## A    INTUITIVE EXAMPLES OF LOW-RANK COMPRESSION

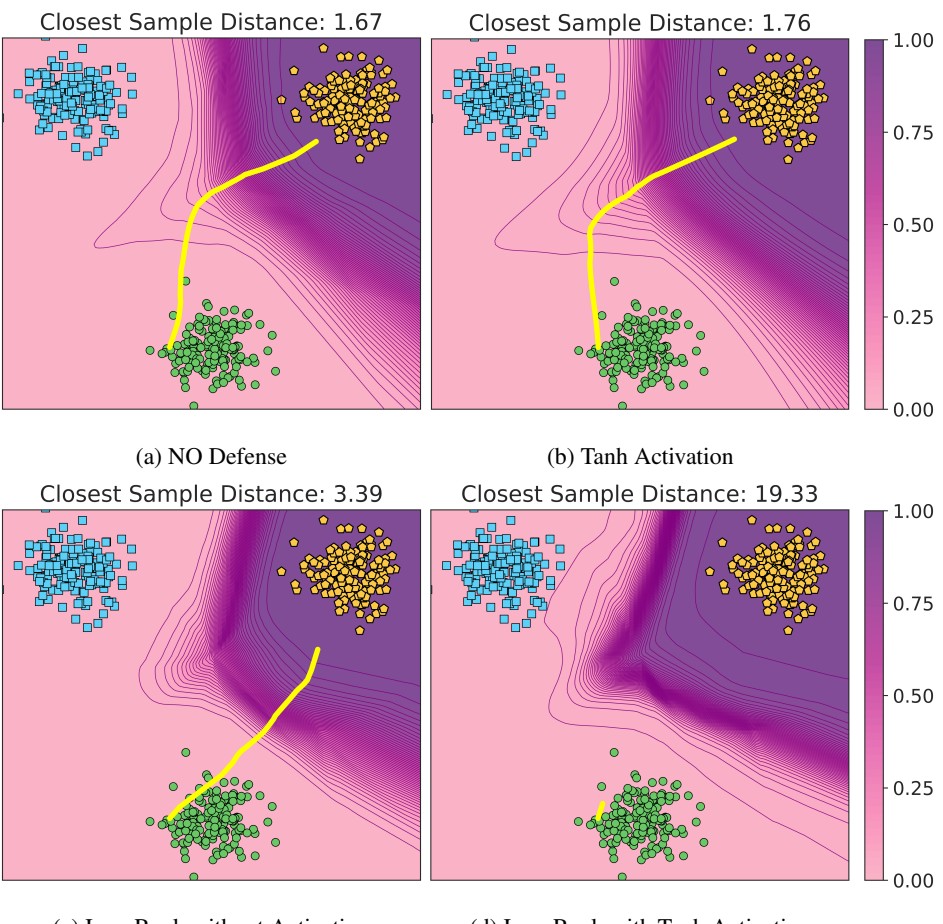

Figure 7: Simple MIA on a 2D toy dataset with three classes. The Background color indicates the models' prediction confidence, and the yellow lines show the intermediate optimization steps of the attack. The optimization starts from a random position, here a sample from the green circle class, and tries to reconstruct a sample from the orange pentagons class. The attack against model without low-rank compression constructs a sample very close to the targeted training data, both with (7b) and without (7a) tanh activation. In the presence of low-rank compression, the model is more inclined to give very high confidence or very low confidence (7c). In this case, the gradient in the space close to the target sample is even smaller, making it difficult to optimize to the range of the target sample. At the same time, this very high or very low output produces a more severe gradient vanishing problem in the presence of tanh activation (7d).

Following Struppek et al. (2024), we present a simple toy example to show how low-rank compression affects model privacy. The example uses a two-dimensional dataset with three classes: blue squares, green circles, and orange pentagons. We first train a three-layer neural network with a hidden dimension of 20 and ReLU activation. This is our baseline model without any defense. Then, we create another model by replacing the last ReLU activation with a Tanh activation. We also create low-rank models with a rank of 2, both with and without the Tanh activation, following the method in Sec. 3.3. In Fig. 7, we show the model confidences across the input space. The models with low-rank compression (Figs. 7c and 7d) tend to give either very low or very high confidence predictions.

We perform a simple MIA on these four models. We start with a random point, specifically a sample from the class of green circles, and optimize it to maximize the model's confidence for the class of orange pentagons. The optimization runs for 2500 steps. The goal is to reveal the features of the

target class, which are just the coordinates of the training samples in this case. Without low-rank compression, the models, with tanh (Fig. 7b) or without it (Fig. 7a), produce similar reconstructed $l_2$ distances: 1.76 and 1.67, respectively. But with low-rank compression, the attacker's optimization results are farther from the target sample, as seen in Fig. 7c, where the distance increases to 3.39. Additionally, using Tanh activation in the low-rank model cause gradient vanishing, as shown in Fig. 7d. This make it much harder for the attacker to optimize, resulting in a final distance of 19.33 away from the target sample. This indicates a significant reduction in the effectiveness of the attack.

## B  CONVERGENCE VALUE OF THE CONFIDENCE ADAPTATION LOSS

For simplicity of notation, we let $t = \hat{y}_c \in (0, 1]$. The confidence adaptation loss is defined as:

$$\mathcal{L} = at^b \log t, \tag{3}$$

where $a$ and $b$ are constants.

**Proposition 1.** *For the function $\mathcal{L} = at^b \log t$, where $a > 0$ and $b > 0$, there exists a minimum point at $t = \exp(-1/b)$ for $t \in (0, 1]$.*

*Proof.* To find the extremum of $\mathcal{L}$, we first compute its derivative with respect to $t$:

$$
\begin{aligned}
\frac{\mathrm{d}\mathcal{L}}{\mathrm{d}t} &= a \left( \frac{\mathrm{d}}{\mathrm{d}t} t^b \log t \right) \\
&= a \left( bt^{b-1} \log t + t^{b-1} \right) \\
&= at^{b-1} \left( b \log t + 1 \right).
\end{aligned} \tag{4}
$$

Setting $\frac{\mathrm{d}\mathcal{L}}{\mathrm{d}t} = 0$ to find the extremum, we obtain:

$$b \log t + 1 = 0. \tag{5}$$

Solving for $t$, we get:

$$t = \exp\left( -\frac{1}{b} \right). \tag{6}$$

Next, we compute the second derivative of $\mathcal{L}$ with respect to $t$ to determine the nature of this extremum:

$$
\begin{aligned}
\frac{\mathrm{d}^2\mathcal{L}}{\mathrm{d}t^2} &= a\frac{\mathrm{d}}{\mathrm{d}t} \left( t^{b-1} \left( b \log t + 1 \right) \right) \\
&= a \left[ (b-1)t^{b-2} \left( b \log t + 1 \right) + t^{b-2} \cdot b \right] \\
&= at^{b-2} \left[ (b-1)(b \log t + 1) + b \right] \\
&= at^{b-2} \left[ b(b-1) \log t + b - 1 + b \right] \\
&= at^{b-2} \left[ b(b-1) \log t + 2b - 1 \right].
\end{aligned} \tag{7}
$$

Substituting $t = \exp\left( -\frac{1}{b} \right)$ into the second derivative:

$$
\begin{aligned}
\left. \frac{\mathrm{d}^2\mathcal{L}}{\mathrm{d}t^2} \right|_{t=\exp\left(-\frac{1}{b}\right)} &= a \exp\left( \frac{2-b}{b} \right) \left[ b(b-1) \log \left( \exp\left( -\frac{1}{b} \right) \right) + 2b - 1 \right] \\
&= a \exp\left( \frac{2-b}{b} \right) \left[ b(b-1) \left( -\frac{1}{b} \right) + 2b - 1 \right] \\
&= a \exp\left( \frac{2-b}{b} \right) \left[ -(b-1) + 2b - 1 \right] \\
&= ab \exp\left( \frac{2-b}{b} \right).
\end{aligned} \tag{8}
$$

Since $a > 0$, $b > 0$, and $t \in (0, 1)$, the second derivative $\frac{\mathrm{d}^2\mathcal{L}}{\mathrm{d}t^2}$ is positive. This indicates that the critical point $t = \exp\left( -\frac{1}{b} \right)$ is indeed a minimum.

Thus, the value $t = \exp\left( -\frac{1}{b} \right)$ is where the confidence adaptation loss $\mathcal{L}$ attains its minimum. $\quad\square$

Based on proposition 1, the coverage value of our confidence adaptation loss is $\exp\left(-\frac{1}{b}\right)$.

## C  SUMMARY OF PREVIOUS DEFENSE METHODS

In this section, we briefly summarize previous defense methods and introduce how they defend against MIAs.

### C.1  MI OVERFITTING

MID (Wang et al., 2021b), BiDO (Peng et al., 2022) and TL (Ho et al., 2024) mainly defend attacks by reducing information encoded in the model output, making attackers harder to overcome MI overfitting issue via getting information from model outputs.

**Mutual Information Regularization based Defense (MID).**  Wang et al. (2021b) enhances model robustness by reducing the mutual information between the model's input and output. The formulation of the loss is as follows:

$$\mathcal{L}_{MID} = \mathcal{L}_{CE} + \lambda I(X, \hat{Y}), \tag{9}$$

where $\lambda$ is a hyperparameter and $I(X, \hat{Y})$ denotes the mutual information between inputs and outputs. However, this mutual information loss term cannot be calculated directly. To overcome this, they apply a variational approach (Alemi et al., 2016) to approximate the mutual information loss term. In practice, this involves Gaussian sampling at the output of the penultimate layer, similar to the approach used in variational autoencoders (VAEs) (Kingma, 2013). The loss functions are as follows:

$$\mathcal{L}_{MID} = \mathcal{L}_{CE} + \lambda(-\frac{1}{2}(1 + \log \sigma^2 - \mu^2 - \sigma^2)), \tag{10}$$

where $\lambda$ is a hyperparameter, $\mu$ and $\sigma$ are the output of the penultimate layer to execute the Gaussian sampling.

**Bilateral Dependency Optimization (BiDO).**  Peng et al. (2022) propose to reduce the dependency $d(\cdot, \cdot)$ between the model inputs $X$ and the intermediate feature $Z$ and improve that between $Z$ and labels $Y$. The loss function is:

$$\mathcal{L}_{BiDO} = \mathcal{L}_{CE} + \lambda_{iz} \sum_{i=1}^{M} d(X, Z_i) - \lambda_{oz} \sum_{i=1}^{M} d(Z_i, Y), \tag{11}$$

where $\lambda_{iz}$ and $\lambda_{oz}$ are hyperparameters. $Z_i, i \in [1, 2, \ldots M]$ represents different intermediate output by the model. In our experiment, we set $M = 3$. The dependency measure $d(\cdot, \cdot)$ can be represented by Constrained Covariance (COCO) (Gretton et al., 2005b) or the Hilbert-Schmidt Independence Criterion (HSIC) (Gretton et al., 2005a). According to the original paper, the HSIC criterion demonstrates superior performance in defense. Therefore, we utilize the HSIC criterion in our experiment.

**Transfer Learning (TL).**  Ho et al. (2024) analyse the fisher information of each layers in classification and inversion tasks. Through the analysis of fisher information between the parameters and different tasks, they find that the previous layers in the model are more important for the model inversion task and the last some layers makes more contributions to the classification task. Therefore, to migrate the effect of MIAs, they pre-train the model on public datasets and freeze the parameters in previous layers when fine-tuning with private datasets. The hyperparameter is the freezing ratio, denoted as $\lambda$.

### C.2  OPTIMIZATION

**Label Smoothing (LS).**  Struppek et al. (2024) explore the effect of the label smoothing technique to MIAs. The loss function of LS is:

$$\mathcal{L}_{LS} = (1 - \lambda)\mathcal{L}_{CE}(\mathbf{y}, \hat{\mathbf{y}}) + \frac{\lambda}{C}\mathcal{L}_{CE}(\mathbf{1}, \hat{\mathbf{y}}), \tag{12}$$

where $\lambda$ is the label smoothing factor, $\mathbf{y}$ is the label, $\hat{\mathbf{y}}$ is the model prediction and $\mathbf{1}$ denotes a vector of length $C$ with all entries set to 1. $C$ is the number of the classes. The factor $\lambda > 0$ will facilitate the inversion attack, while $\lambda < 0$ will have a strong defensive effect.

LS with negative label smoothing factor can effectively impede the optimization process of the attacks. To assess this, we select some challenging samples as target identities and apply the IF attacks on a ResNet-152 model pre-trained with various public datasets, recording the confidence levels predicted by the victim classifier at each optimization step. The results are shown in Fig. 8. When the model is pre-trained on ImageNet, we observe that the target confidence remained consistently low, indicating that the optimization process is significantly obstructed. However, when the model is pre-trained on MS-Celeb-1M, the obstruction effect diminish, making it more challenging to defend against the attack in this scenario. We hypothesize that the distinct distributions between ImageNet and FaceScrub datasets necessitate substantial adjustment of model parameters during training, allowing LS to serve as an effective defense. Conversely, because MS-Celeb-1M and FaceScrub both consist of human faces , the required parameter adjustments are minimal, which may reduce the effectiveness of LS in this context.

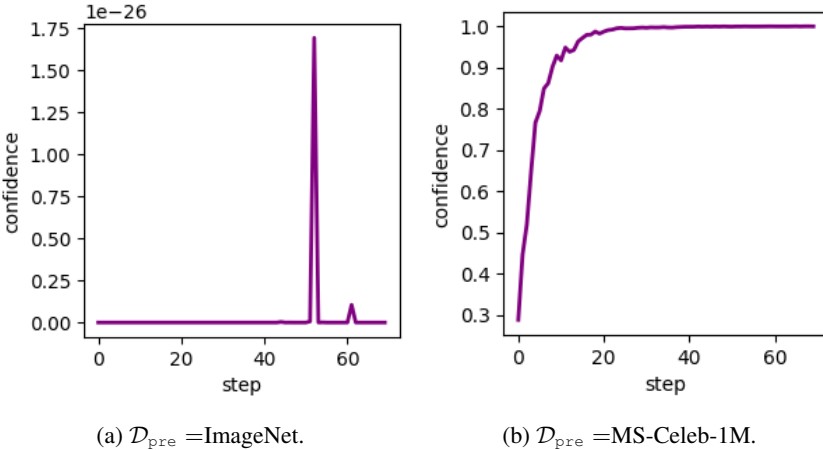

(a) $\mathcal{D}_{\text{pre}}$ =ImageNet.          (b) $\mathcal{D}_{\text{pre}}$ =MS-Celeb-1M.

Figure 8: The trend of classifier confidence under IF attack across different pre-trained datasets $\mathcal{D}_{\text{pre}}$ with the LS defense method.

## D  EXPERIMENT DETAILS

### D.1  CLASSIFIER TRAINING

We train the classifier models based on prior research, incorporating both the low-resolution scenario (Zhang et al., 2020) and the high-resolution scenario (Struppek et al., 2022).

Following previous researches (Zhang et al., 2020; Struppek et al., 2022), the MS-Celeb-1M backbones in the low-resolution are provided by Face.evoLVe (Wang et al., 2021a), and the ImageNet backbones in the high-resolution scenarios are the default pre-trained weights of torchvision models (maintainers & contributors, 2016). For the MS-Celeb-1M backbones in high-resolution scenarios, we pre-train the models on the whole MS-Celeb-1M dataset for 5 epochs.

In low-resolution scenarios, we use the SGD optimizer (Sutskever et al., 2013) with an initial learning rate of 0.01 and momentum of 0.9. The batch size is set to 128. All data samples are cropped and resized to $64 \times 64$ for IR-152 and $112 \times 112$ for FaceNet-112, with a random horizontal flip applied with a probability of 50%.

For high-resolution scenarios, the Adam optimizer (Diederik, 2014) is employed with an initial learning rate of 0.001 and $\beta = (0.9, 0.999)$. The batch size is 96. All data samples are normalized with $\mu = 0.5$ and $\sigma = 0.5$, and resized to $224 \times 224$. The training samples are augmented through random cropping within a scale range of $[0.85, 1.0]$ and a fixed aspect ratio of 1.0. Crops are resized

back to $224 \times 224$, followed by random color jittering with brightness and contrast factors of $0.2$, and saturation and hue factors of $0.1$. A horizontal flip is applied with a probability of $50\%$.

In both scenarios, the models are trained for $100$ epochs, with the learning rate reduced by a factor of $0.1$ after epochs $75$ and $90$.

For the target classifier, to ensure a fair comparison of different defense algorithms, we meticulously tuned the hyperparameters of each algorithm to maintain nearly identical classification performance across models. The hyperparameters of previous defense methods are described in Appendix C. For our CALoR, we set $a = 1$ and $b = 8$ in the confidence adaptation loss, and the adjustable hyperparameter is the rank $r$ in the low-rank compression. The specific hyperparameters used in the main paper are presented in Table 6. In high-resolution scenarios with ImageNet as pre-training dataset, to reduce the impact of Tanh's vanishing gradient effect on training, we first trained for $50$ epochs without defense, and then replaced the classification head with the low-rank classification head.

For the evaluation models, we use FaceNet-112 and MaxViT in low- and high-resolution scenarios, respectively. The classification accuracy in the test dataset are shown in Table 7.

Table 6: Hyperparameters for the experimental settings in the main paper. The models are trained on the FaceScrub dataset, with the IR-152 architecture used in the low-resolution scenario and ResNet-152 in the high-resolution scenario. The settings in the table are as follows: (A) Low-resolution scenario with MS-Celeb-1M as the pre-training dataset, (B) High-resolution scenario with MS-Celeb-1M as the pre-training dataset, and (C) High-resolution scenario with ImageNet as the pre-training dataset.

| Settings | A | B | C |
|---|---|---|---|
| **MID** | 0.1 | 0.005 | 0.005 |
| **BiDO** | 0.01, 0.1 | 0.15, 1.5 | 0.05, 0.5 |
| **LS** | $-0.3$ | $-0.01$ | $-0.02$ |
| **TL** | 50% | 70% | 60% |
| **CALoR** | 30 | 50 | 32 |

Table 7: The test classification accuracy of the evaluation models on different private datasets used in our experiments. FaceNet-112 is utilized for evaluation in the low-resolution scenario, while MaxViT is employed for the high-resolution scenario.

| Test Acc | FaceScrub | CelebA |
|---|---|---|
| **FaceNet-**112 | 99.38% | 95.88% |
| **MaxViT** | 99.41% | 97.23% |

## D.2 ATTACKS

We perform various kinds of MI attacks, including GMI (Zhang et al., 2020), KED (Chen et al., 2021), Mirror (An et al., 2022), PPA (Struppek et al., 2022), LOMMA (Nguyen et al., 2023), PLG (Yuan et al., 2023) and IF (Qiu et al., 2024). Following previous research (Zhang et al., 2020; Yuan et al., 2023), we reconstruct $5$ images per class. Due to the high time costs for MIAs, we performed attacks on the first $100$ classes.

In our experimental setup, the prior dataset used by attackers is FFHQ, in line with previous researches (Struppek et al., 2024; Ho et al., 2024). Attackers leverage this dataset to train GANs or surrogate classifiers (Nguyen et al., 2023). Specifically, GMI, KED, LOMMA, and PLG train customized GANs, while Mirror, PPA, and IF utilize a pre-trained FFHQ StyleGAN2-Ada model provided by Karras et al. (2020).

## E ADDITIONAL EXPERIMENT RESULTS

In this section, we present some additional experimental results.

**More attack methods.** We evaluate GMI (Zhang et al., 2020), KED (Chen et al., 2021), LOMMA (Nguyen et al., 2023), Mirror (An et al., 2022) and PPA (Struppek et al., 2022) in low-resolution scenarios against IR-152. The results are presented in Table 8, 9 and 10. The results show that our defense is significantly more effective than previous methods in most scenarios.

**Evaluation on the CelebA dataset.** We evaluated the defense methods on the CelebA dataset under both low and high-resolution scenarios. The results, presented in Table 11, 12 and 14, indicate that our method consistently outperforms all other defense algorithms. This demonstrates the robustness of our approach across different datasets and resolutions.

**Evaluation on transformer-based classifiers.** In the previous experiments, we analyzed the impact of defense algorithms on convolution-based classifiers. To further explore this, we extended our experiments to include transformer-based models. The attack results in high-resolution scenarios are shown in Table 13 and 14. The results reveal that while the defense algorithm is generally less effective for transformer-based model structures, our method continues to achieve SOTA results.

Table 8: GMI and KED attack results against IR-152 models trained on FaceScrub dataset in the low-resolution scenario. The models are pre-trained with MS-Celeb-1M.

| Method | Test Acc | GMI | | | | KED | | | |
|---|---|---|---|---|---|---|---|---|---|
| | | ↓ Acc@1 | ↓ Acc@5 | ↑ $\delta_{eval}$ | ↑ $\delta_{face}$ | ↓ Acc@1 | ↓ Acc@5 | ↑ $\delta_{eval}$ | ↑ $\delta_{face}$ |
| **NO Defense** | 98.2% | 8.0% | 11.6% | 2466.46 | 1.45 | 37.8% | 52.6% | 2113.85 | 1.04 |
| **MID** | 97.0% | 11.2% | 20.6% | 2507.09 | 1.32 | 60.0% | 77.6% | 1846.08 | 0.81 |
| **BiDO** | 95.2% | 3.6% | 12.6% | 2390.00 | 1.22 | 20.2% | 40.6% | 2257.25 | 0.98 |
| **LS** | 97.3% | 9.2% | 20.6% | 2481.68 | 1.36 | 26.8% | 48.6% | 2345.70 | 1.12 |
| **TL** | 95.4% | **1.8%** | **5.0%** | **2694.00** | 1.46 | 21.8% | 33.8% | 2279.65 | 1.06 |
| **CALoR(ours)** | 97.0% | 3.2% | 5.8% | 2659.06 | **1.54** | **9.4%** | **20.4%** | **2532.08** | **1.33** |

Table 9: Mirror and PPA attack results against IR-152 models trained on FaceScrub dataset in the low-resolution scenario. The models are pre-trained with MS-Celeb-1M.

| Method | Test Acc | Mirror | | | | PPA | | | |
|---|---|---|---|---|---|---|---|---|---|
| | | ↓ Acc@1 | ↓ Acc@5 | ↑ $\delta_{eval}$ | ↑ $\delta_{face}$ | ↓ Acc@1 | ↓ Acc@5 | ↑ $\delta_{eval}$ | ↑ $\delta_{face}$ |
| **NO Defense** | 98.2% | 40.4% | 57.4% | 1911.03 | 0.95 | 91.2% | 95.2% | 1203.95 | 0.63 |
| **MID** | 97.0% | 45.2% | 59.2% | 1918.89 | 0.97 | 90.4% | 95.0% | 1247.68 | 0.65 |
| **BiDO** | 95.2% | 13.8% | 27.8% | 2271.02 | 1.06 | 53.6% | 77.2% | 1708.76 | 0.78 |
| **LS** | 97.3% | 35.6% | 58.4% | 1937.67 | 0.98 | 91.6% | 94.4% | 1246.78 | 0.66 |
| **TL** | 95.4% | 14.4% | 29.4% | 2231.25 | 1.13 | 57.0% | 77.0% | 1626.81 | 0.78 |
| **CALoR(ours)** | 97.0% | **6.8%** | **15.2%** | **2487.72** | **1.37** | **24.4%** | **41.2%** | **2123.30** | **1.13** |

Table 10: LOMMA attack results against IR-152 models trained on FaceScrub dataset in the low-resolution scenario. The models are pre-trained with MS-Celeb-1M. Note that LOMMA is a plug-and-play technique that can seamlessly combine with existing generative model inversion attacks. In our experiments, we adhered to the official configurations, integrating LOMMA with both GMI and KED.

| Method | Test Acc | LOMMA+GMI | | | | LOMMA+KED | | | |
|---|---|---|---|---|---|---|---|---|---|
| | | ↓ Acc@1 | ↓ Acc@5 | ↑ $\delta_{eval}$ | ↑ $\delta_{face}$ | ↓ Acc@1 | ↓ Acc@5 | ↑ $\delta_{eval}$ | ↑ $\delta_{face}$ |
| **NO Defense** | 98.2% | 48.0% | 59.1% | 2063.91 | 1.05 | 75.1% | 87.2% | 1641.22 | 0.72 |
| **MID** | 97.0% | 56.7% | 75.1% | 1879.30 | 0.83 | 66.6% | 86.6% | 1708.48 | 0.72 |
| **BiDO** | 95.2% | 29.8% | 48.6% | 2253.56 | 0.92 | 64.6% | 83.6% | 1782.43 | 0.72 |
| **LS** | 97.3% | 50.8% | 64.9% | 2095.55 | 0.99 | 78.6% | 90.4% | 1526.01 | 0.70 |
| **TL** | 95.4% | **14.9%** | 28.5% | 2387.43 | 1.13 | 38.8% | 58.4% | 2031.39 | 0.91 |
| **CALoR(ours)** | 97.0% | 16.8% | **27.6%** | **2442.43** | **1.23** | **31.8%** | **48.4%** | **2185.88** | **1.08** |

Table 11: IF and PLG attack results against IR-152 models trained on CelebA dataset in the low-resolution scenario. The models are pre-trained with MS-Celeb-1M.

| Method | Test Acc | IF | | | | PLG | | | |
|---|---|---|---|---|---|---|---|---|---|
| | | ↓ Acc@1 | ↓ Acc@5 | ↑ $\delta_{eval}$ | ↑ $\delta_{face}$ | ↓ Acc@1 | ↓ Acc@5 | ↑ $\delta_{eval}$ | ↑ $\delta_{face}$ |
| NO Defense | 91.1% | 34.2% | 49.4% | 1547.73 | 1.16 | 58.0% | 81.8% | 1419.52 | 0.93 |
| MID | 89.2% | 33.8% | 43.6% | 1625.00 | 1.21 | 73.2% | 88.6% | 1307.62 | 0.74 |
| BiDO | 86.1% | 24.6% | 41.2% | 1626.31 | 1.27 | 58.0% | 82.4% | 1435.90 | 0.88 |
| LS | 87.7% | 20.6% | 34.8% | 1681.71 | 1.19 | 91.6% | 99.2% | 1180.62 | 0.70 |
| TL | 86.3% | 6.4% | 13.4% | 1886.00 | **1.44** | 70.4% | 90.8% | 1353.79 | 0.81 |
| CALoR(ours) | 86.3% | **3.6%** | **8.4%** | **1905.50** | 1.41 | **5.4%** | **15.6%** | **1825.39** | **1.34** |

Table 12: IF attack results against ResNet-152 models trained on CelebA dataset in the high-resolution scenario. The models are pre-trained with MS-Celeb-1M.

| Method | Test Acc | IF | | | |
|---|---|---|---|---|---|
| | | ↓ Acc@1 | ↓ Acc@5 | ↑ $\delta_{eval}$ | ↑ $\delta_{face}$ |
| NO Defense | 96.3% | 99.8% | 100.0% | 276.82 | 0.46 |
| MID | 94.3% | 98.6% | 99.8% | 265.71 | 0.44 |
| BiDO | 94.7% | 97.8% | 99.6% | 293.39 | 0.48 |
| LS | 93.6% | 92.6% | 98.8% | 342.43 | 0.61 |
| TL | 94.1% | 100.0% | 100.0% | 229.10 | 0.37 |
| CALoR(ours) | 94.4% | **62.8%** | **79.4%** | **416.74** | **0.80** |

Table 13: IF attack results against Swin-v2 models trained on FaceScrub dataset in the high-resolution scenario. The models are pre-trained with MS-Celeb-1M.

| Method | Test Acc | IF | | | |
|---|---|---|---|---|---|
| | | ↓ Acc@1 | ↓ Acc@5 | ↑ $\delta_{eval}$ | ↑ $\delta_{face}$ |
| NO Defense | 97.8% | 45.6% | 70.2% | 232.24 | 0.66 |
| MID | 97.3% | 18.0% | 40.8% | 282.98 | 0.82 |
| BiDO | 97.9% | 35.2% | 56.2% | 252.90 | 0.64 |
| LS | 97.6% | 29.4% | 53.8% | 248.68 | 0.68 |
| TL | 98.2% | 32.2% | 58.2% | 246.44 | 0.63 |
| CALoR(ours) | 97.5% | **16.0%** | **40.0%** | **300.96** | **0.86** |

Table 14: IF attack results against ViT-B/16 models trained on CelebA dataset in the high-resolution scenario. The models are pre-trained with MS-Celeb-1M.

| Method | Test Acc | IF | | | |
|---|---|---|---|---|---|
| | | ↓ Acc@1 | ↓ Acc@5 | ↑ $\delta_{eval}$ | ↑ $\delta_{face}$ |
| NO Defense | 95.3% | 84.6% | 94.2% | 363.25 | 0.65 |
| MID | 92.6% | 93.8% | 97.0% | 320.57 | 0.56 |
| BiDO | 91.4% | 79.4% | 89.2% | 381.65 | 0.69 |
| LS | 92.2% | 88.6% | 97.0% | 366.45 | 0.62 |
| TL | 95.6% | 94.4% | 97.4% | 326.24 | 0.56 |
| CALoR(ours) | 91.9% | **73.0%** | **87.4%** | **403.60** | **0.75** |

## F MODEL ROBUSTNESS IN OTHER ASPECTS

### F.1 KNOWLEDGE EXTRACTION SCORE OF MIAS

Following the approach of Struppek et al. (2024), we compute the Knowledge Extraction Score (KES), a metric designed to quantify the discriminative information extracted about distinct classes from MIAs. Specifically, we train a surrogate ResNet-152 (He et al., 2016) classifier on the synthetic

data generated by attacks and evaluate its classification accuracy on the target model's original training set. The intuition behind KES is that a more successful inversion attack will enable the surrogate model to better differentiate between classes.

In our experiments, we employ synthetic attack data generated by IF attacks against a ResNet-152 (He et al., 2016) model trained on the FaceScrub dataset. The ResNet-152 model is pre-trained on either the MS-Celeb-1M or ImageNet datasets. The results of these experiments are shown in Table 15. The results shows that the knowledge extraction score of IF attacks against classifiers with our CALoR defense method achieve the lowest results, which indicates that MIAs can hardly extract private information from models with our defense methods.

Table 15: Knowledge extraction scores in high-resolution scenarios. The target models and the student models are pre-trained with MS-Celeb-1M (MS) or ImageNet (IN).

| Method | MS | | IN | |
|---|---|---|---|---|
| | Test Acc | ↓KES | Test Acc | ↓KES |
| NO | 98.5% | 91.8% | 92.2% | 72.7% |
| MID | 96.8% | 90.2% | 88.2% | 83.5% |
| BiDO | 96.3% | 91.6% | 88.6% | 72.8% |
| LS | 96.4% | 90.8% | 88.8% | 68.4% |
| TL | 96.6% | 96.0% | 88.4% | 60.0% |
| **CALoR(ours)** | 96.8% | **62.0%** | 89.4% | **41.8%** |

## F.2 ADVERSARIAL ROBUSTNESS

In this section, we investigate if training with different defense methods has an impact on a model's robustness against adversarial attacks. Following Struppek et al. (2024), we apply the following attacks to test model robustness:

- **Fast Gradient Sign Method (FGSM)** (Goodfellow et al., 2014): One-step white-box attack. $\epsilon = 8/255$.

- **Projected Gradient Descent (PGD)** (Mądry et al., 2017): Multi-step white-box attack. $\epsilon = 8/255$, step size $= 2/255$, steps $= 10$, random start $=$ True.

- **Basic Iterative Method (BIM)** (Kurakin et al., 2018): Multi-step white-box attack. $\epsilon = 8/255$, step size $= 2/255$, steps $= 10$.

- **One Pixel Attack** (Su et al., 2019): Multi-step black-box attack. Pixels $= 1$, steps $= 10$, population size $= 10$.

Adversarial attacks are conducted on test samples that are excluded from the training data. These attacks are evaluated in both targeted and untargeted settings. In the untargeted scenario, an attack is considered successful if the predicted label differs from the ground truth label. For targeted attacks, the target label is set to the original label plus one, and the attack is successful if the model predicts this target label. In both cases, we measure the attack success rate (ASR), where a lower ASR indicates greater robustness of the model to adversarial perturbations. The results in Table 16 demonstrate that training a model with our defense method can make a model more robust to adversarial examples, especially in the targeted scenario.

## F.3 BACKDOOR ROBUSTNESS

In addition to adversarial robustness, we also investigate of training with our CALoR has an impact on backdoor attacks. Due to the relatively small number of samples per class in FaceScrub and CelebA, it is not able to stably train the models (Struppek et al., 2024). Therefore, we train ResNet-152 models on a poisoned ImageNette (fastai, 2022) dataset, which is a subset of ImageNet (Deng et al., 2009) classes. Following the settings of Struppek et al. (2024), we evaluate the backdoor robustness on the following attack methods:

Table 16: Adversarial attack results against ResNet-152 models trained on FaceScrub dataset in the high-resolution scenario. $\mathcal{D}_{pre}$ means the public dataset to pre-train the model, incliding MS-Celeb-1M (MS) and ImageNet (IN).

| $\mathcal{D}_{pre}$ | Method | Test Acc | Untargeted Attacks | | | | Targeted Attacks | | | |
|---|---|---|---|---|---|---|---|---|---|---|
| | | | ↓ FGSM | ↓ PGD | ↓ BIM | ↓ OnePixel | ↓ FGSM | ↓ PGD | ↓ BIM | ↓ OnePixel |
| MS | NO Defense | 98.5% | 98.3% | 100.0% | 100.0% | **1.2%** | 72.4% | 100.0% | 100.0% | 0.0% |
| | MID | 96.8% | 99.8% | 100.0% | 100.0% | 3.2% | 52.4% | 100.0% | 100.0% | 0.0% |
| | BiDO | 96.3% | 97.8% | 100.0% | 100.0% | 4.1% | 56.6% | 100.0% | 100.0% | 0.0% |
| | LS | 96.4% | 99.3% | 100.0% | 100.0% | 4.1% | 52.7% | 100.0% | 100.0% | 0.0% |
| | TL | 96.6% | 99.3% | 100.0% | 100.0% | 2.4% | 63.9% | 100.0% | 100.0% | 0.0% |
| | LoR(ours) | 96.6% | 88.3% | 100.0% | 100.0% | 2.7% | 2.2% | **87.3%** | 91.5% | 0.0% |
| | CALoR(ours) | 96.8% | **85.9%** | 100.0% | 100.0% | 2.4% | **2.0%** | 89.0% | **91.0%** | 0.0% |
| IN | NO Defense | 92.2% | 98.3% | 100.0% | 100.0% | 11.5% | 49.8% | 100.0% | 100.0% | 0.5% |
| | MID | 88.2% | 91.5% | 100.0% | 100.0% | 12.7% | 3.7% | 92.4% | 94.4% | **0.0%** |
| | BiDO | 88.6% | 98.0% | 100.0% | 100.0% | **9.5%** | 32.7% | 100.0% | 100.0% | 0.2% |
| | LS | 88.8% | 20.0% | 21.2% | 20.5% | 12.0% | 27.6% | 98.8% | 98.8% | **0.0%** |
| | TL | 88.4% | 85.1% | 100.0% | 100.0% | 12.9% | 3.7% | 97.3% | 96.1% | 0.2% |
| | CALoR(ours) | 89.4% | 96.3% | 100.0% | 100.0% | 9.8% | **0.7%** | 69.5% | 69.8% | **0.0%** |

- **BadNets** (Gu et al., 2017): We add a $9 \times 9$ checkerboard pattern to the lower right corner of reach image. In total, $1\%$ of all images are poisioned and labeled as class $0$.

- **Blended** (Chen et al., 2017): We interpolate each poisoned image with a fixed Gaussian noise pattern. The blend ratio is set to 0.1. In total, $1\%$ of all images are poisoned and labeled as class $0$.

For evaluation, we computed the classification accuracy on the test splits. To calculate the attack success rate, all test images are added with triggers. An poisoned image is viewed success if it is classified as the class $0$. Lower attack success rate means the more robust the model is to the backdoor attack.

Table 17: Backdoor attack results against ResNet-152 models trained on the ImageNette dataset.

| Trigger | Defense | ↑Clean Accuracy | ↓ Attack Success Rate |
|---|---|---|---|
| Clean | NO defense | 83.1% | - |
| | LoR | 80.5% | - |
| | CALoR | 81.0% | - |
| BadNets | NO defense | 79.7% | 89.9% |
| | LoR | 82.4% | 88.6% |
| | CALoR | 82.7% | **88.2%** |
| Blended | NO defense | 85.6% | 87.4% |
| | LoR | 77.9% | **43.9%** |
| | CALoR | 76.4% | 47.3% |

## G THE ATTACK RESULTS OF DIFFERENT RANKS IN HIGH-RESOLUTION SCENARIOS.

Table 18 presents the IF attack results on models with varying ranks. These models are pre-trained on the MS-Celeb-1M dataset and fine-tuned on the FaceScrub dataset in the low-resolution scenario. The results demonstrate that a low rank is sufficient to achieve high model performance, while increasing the rank leads to a higher risk of privacy leakage.

Table 18: IF attack results against ResNet-152 models with different ranks in the classifier head in high-resolution scenarios.

| Rank | Test Acc | ↓ Acc@1 | ↓ Acc@5 | ↑ $\delta_{eval}$ | ↑ $\delta_{face}$ |
|------|----------|---------|---------|---------|----------|
| 20 | 92.8% | 18.6% | 33.2% | 581.11 | 1.20 |
| 35 | 95.5% | 39.2% | 60.6% | 498.34 | 1.01 |
| 40 | 96.1% | 51.2% | 69.4% | 480.28 | 0.97 |
| 50 | 96.6% | 64.8% | 83.2% | 431.29 | 0.86 |
| 75 | 97.5% | 80.4% | 91.4% | 392.00 | 0.76 |
| 100 | 97.7% | 93.0% | 97.6% | 349.14 | 0.66 |
| 150 | 98.6% | 97.8% | 99.4% | 321.47 | 0.60 |
| 200 | 98.7% | 98.8% | 100.0% | 308.44 | 0.58 |
| 300 | 98.9% | 98.2% | 99.2% | 303.51 | 0.56 |
| 500 | 98.9% | 99.0% | 100.0% | 303.45 | 0.55 |

## H  MORE VISUALIZATION RESULTS

In this section, we present additional visual examples from the main experiments. Figure 9 shows reconstructed samples from PLG attacks on ResNet-152 under various defense methods in high-resolution scenarios. Meanwhile, Figure 10 displays reconstructed samples from IF attacks on IR-152 in low-resolution scenarios. The FaceScrub dataset is used as the private dataset in these experiments.

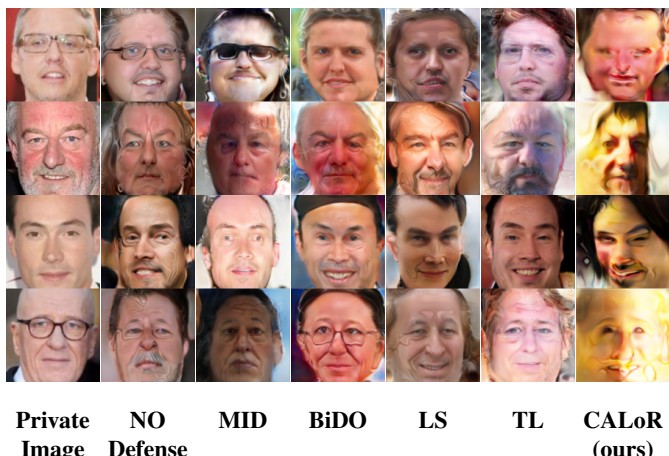

| Private
Image | NO
Defense | MID | BiDO | LS | TL | CALoR
(ours) |
|---|---|---|---|---|---|---|

Figure 9: Visual comparison of PLG attacks against ResNet-152 under different defense strategies.

## I  EXPERIMENTS OF MOTIVATIONS

### I.1  EXPERIMENTS OF CONFIDENCE AND ATTACK ACCURACY

In order to explore the relationship between model prediction confidence and attack accuracy, we train a series of classifiers with different level of average confidence of private images. The model is IR-152 trained on the FaceScrub dataset in the low-resolution scenario. In practice, we train a model and get the model weight in different periods of the training process, thereby getting many models with different level of confidence.

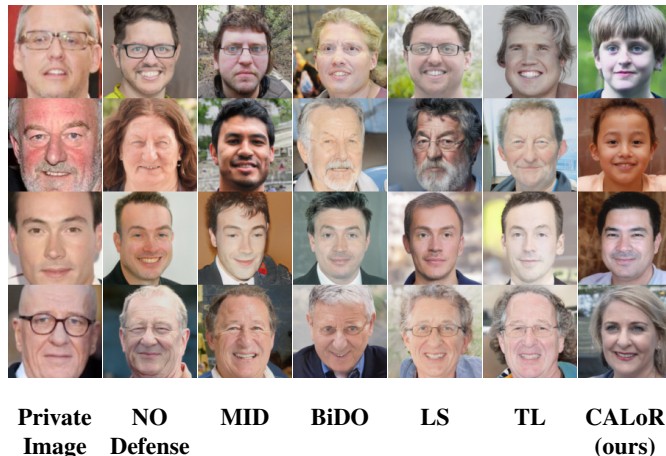

**Private NO MID BiDO LS TL CALoR**
**Image Defense (ours)**

Figure 10: Visual comparison of IF attacks against IR-152 under different defense strategies in the low-resolution scenario.

### I.2 EXPERIMENTS OF AUTOENCODERS FOR LOW-RANK COMPRESSION

**Autoencoder Architecture.** We train multiple autoencoders with different ranks $r$, where $r \in \{2, 4, 8, 16, 32, 64, 128, 256, 512\}$. We decompose the autoencoder into an encoder $\mathcal{E}$ and a decoder $\mathcal{D}$. $\mathcal{E}$ transform the input image into a $r$-dimension feature vector, and $\mathcal{D}$ recover the feature vector into the origin images. The encoder use the backbone of ResNet-50 (He et al., 2016) pre-trained on ImageNet (Deng et al., 2009), and the decoder is the same as the architecture of generator provided by GMI (Zhang et al., 2020).

**Training.** We train the autoencoders on the train split of the CelebA dataset (Liu et al., 2015). All training samples are augmented through random cropping within a scale range of $[0.85, 1.0]$ and a fixed aspect ratio of $1.0$. Crops are resized back to $224 \times 224$, followed by random color jittering with brightness and contrast factors of $0.2$, and saturation and hue factors of $0.1$. A horizontal flip is applied with a probability of $50\%$. The batch size is $32$. The autoencoders are trained for $100$ epochs, with Adam (Diederik, 2014) optimizers with initial learning rate of $0.0001$ and $\beta = (0.9, 0.999)$. The learning rate is reduced by a factor of $0.1$ after epochs $50, 75$ and $90$.

**Evaluation.** We evaluate the autoencoders on the test split of the CelebA dataset. We compute the average MSE loss between the origin images and reconstructed samples outputed by the autoencoders. Moreover, we use a MaxViT (Tu et al., 2022) pre-train on CelebA dataset to re-classify the reconstructed samples and calculate the accuracy. The test accuracy of the MaxViT is $96.44\%$.

## J BACK-PROPAGATION FUNCTIONS OF SOME COMMON NON-LINEAR ACTIVATIONS.

Fig. 11 shows the back-propagation functions of some common non-linear activations, including ReLU, Sigmoid and Tanh. It shows that when the independent variable $x$ gets larger, the gradient of ReLU maintain at a very high level, and that of Tanh and Sigmoid gradually converging to $0$. Tanh converges to $0$ faster, meaning that it is more likely to cause the gradient vanishing problem.

## K IMAGE CATEGORY IN THE INVERSION SPACE

Since the confidence of the target model's prediction can never fully reach $1$, we consider images with confidence greater than $0.999$ as belonging to the inversion space. Figure 12 provides some visual examples. A success sample is correctly classified by another classifier, indicating that it exposes private features, while adversarial and noisy samples do not. The adversarial sample exists within the attacker's search space, meaning it can be generated by the attacker. In contrast, a noisy

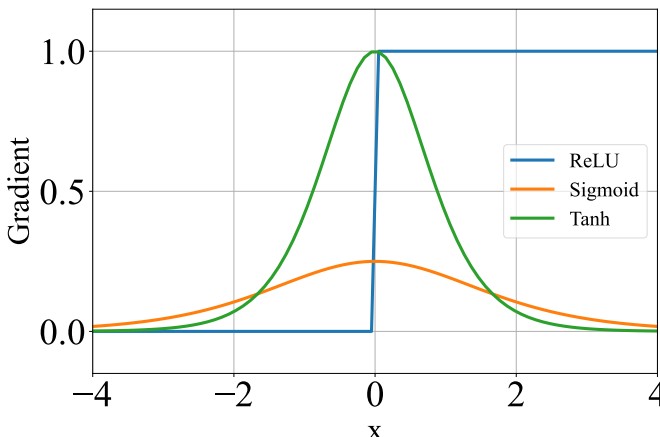

Figure 11: Back-propagation functions of some common non-linear activations.

sample falls outside the attacker's search space due to the attacker's prior knowledge. Both types of samples indicate that the model is overfitted to the target, lacks generalization, and is limited in its ability to reveal privacy-sensitive features.

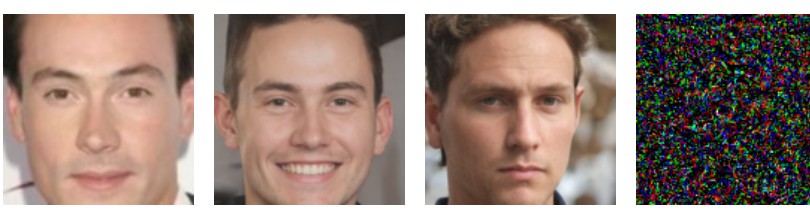

Private Image     Success Sample   Adversarial Sample    Noisy Sample

Figure 12: Images in the inversion space.

## L  VIEWING LOW-RANK COMPRESSION FROM ANOTHER PERSPECTIVE

In this section, we consider the low-rank compression from another perspective. It enlarge the inversion space so that the proportion of success space in the inversion space will decrease.

Table 19: The rank of the weight matrix in the linear classifier head. $m$ and $n$ means the dimensionality of its input and output. The experiment setting is the same as Table 1.

|  | No Defense | MID | BiDO | LS | TL | CALoR($\mathcal{C}_A$) | CALoR($\mathcal{C}_B$) |
|---|---|---|---|---|---|---|---|
| $m$ | 512 | 256 | 512 | 512 | 512 | 512 | 30 |
| $n$ | 530 | 530 | 530 | 530 | 530 | 30 | 530 |
| rank | 512 | 256 | 512 | 512 | 512 | 30 | 30 |

Consider a model $f_\theta(\mathbf{x}) = \mathbf{W}\mathbf{x}$, where $\mathbf{x} \in \mathbb{R}^m$ and $\mathbf{W} \in \mathbb{R}^{n \times m}$. The attacker's goal is to invert the original input $\mathbf{x}$ from the model output $\mathbf{y} = f_\theta(\mathbf{x})$. According to the rank-nullity theorem, the dimension of the null space of $\mathbf{W}$ is given by:

$$\text{Null}(\mathbf{W}) = m - \text{rank}(\mathbf{W}). \tag{13}$$

This implies that if a solution exists, the dimension of the free variables in the solution for $\mathbf{x}$ is $d = m - \text{rank}(\mathbf{W})$. Therefore, as the $\text{rank}(\mathbf{W})$ decreases, the dimension $d$ increases, which

enlarges the inversion space. Consequently, it becomes more difficult for attackers to recover the original input. In our approach, we decompose the weight matrix $\mathbf{W}$ of the classifier into the product of two low-rank matrices, specifically $\mathbf{W}^{n \times m} = \mathbf{W}_B^{n \times r} \mathbf{W}_A^{r \times m}$, where $r \ll m, n$. Thus, $\text{rank}(\mathbf{W}) \leq \min\{\text{rank}(\mathbf{W}_A), \text{rank}(\mathbf{W}_B)\} \leq r$. We analyse the classification header weights of various models trained on different datasets and calculate their weight matrices with the NumPy API function `np.linalg.matrix_rank` (Harris et al., 2020). Part of the result is shown in Table 19. They show that these weight matrices are typically full-rank. Therefore, in the absence of low-rank compression, $\text{rank}(\mathbf{W}) = \min(m, n)$. When low-rank compression is applied, $\text{rank}(\mathbf{W}) = r$. Since $r$ is significantly smaller than $\min(m, n)$, our method substantially increases the dimensionality of the inversion space. Therefore, the proportion of success space in the inversion space decrease, making the attackers more difficult to execute success attacks.

## M    IMPACT, LIMITATION AND FUTURE WORK

Deep learning holds tremendous potential across various domains. However, its applications must be secure and protect user privacy, which can be threatened by MIAs. Our research find that when models have strong performance, existing defense methods are insufficient to resist the latest attack methods. Notably, our comprehensive research significantly increase the difficulty of MIAs. Therefore, we recommend integrating comprehensive defense strategies when fine-tuning pre-trained models for downstream tasks involving sensitive or private data.

Currently, our research, like most MIA-related studies, focuses on classification tasks. In the future, it will be important to study other tasks such as object detection and image segmentation. Furthermore, with the rise of multimodal large models, it is necessary to explore how these models handle privacy concerns. Investigating the security and privacy challenges in multimodal systems will be essential as deep learning continues to expand into real-world applications.

