# OpenReview forum: "CALoR: Towards Comprehensive Model Inversion Defense"
_ICLR.cc/2025/Conference — ICLR 2025 Conference Withdrawn Submission_

### Official Review · Reviewer_roMB · 2024-10-28

**Soundness:** 1
**Presentation:** 1
**Contribution:** 1
**Rating:** 3
**Confidence:** 4

**Summary:**

This paper proposes a defense method named CALoR against model inversion attacks. There are two key components of the proposed method. One is biasing the attack optimization target through confidence adaptation and the other is low-rank compression to mitigate the privacy leakage.

**Strengths:**

1. The model inversion attacks are very important and the privacy concern of the (private) training data should be carefully handled.

2. The overall empirical performance compared with other existing defense work is impressive.

**Weaknesses:**

1. The presentation of this work is extremely poor which makes it look like an incomplete paper, especially for the methodology part. For example, after reading Section 3.2, I still have no idea about how to ensure the utility of the classification model after executing the second stage. In Section 3.3, the authors talk about the latent vector z but without further discussing why z is relevant to their method. Lots of main details are missing/require the audience to refer to the appendix. I suggest the authors revise the manuscript significantly to make it more clear.

2. The motivation of Eqn. (2) is not clear. Through the simple calculation of the derivative, we can see that the final objective is to ensure that \hat{y}_c should converge to exp(-1/b). Why not directly use the MSE loss to mislead the attack objective? Additionally, the hyperparameter a in Eqn. (2) does not play much role.

3. Low-rank compression is a common technique to cause information loss which can be further utilized to protect private input data. I would suggest the authors weaken their claim about the contribution to this part and shorten the relevant paragraphs.

4. This work lacks theoretical analysis about the performance decrease bound due to confidence adaptation and low-rank compression which puts the utility of CALoR into question.

5. I would suggest the authors report the uncertainty (e.g., standard deviation) of the experiment results as well.

**Questions:**

1. Why the hyperparameter a is needed in Eqn. (2)? I think it can be incorporated into the step size.

2. How to properly set the hyperparameter b? Any theoretical justification?

3. In Table 4, a rank of 20 also shows a good performance. Why highlight the rank of 30 in the main text?

4. How to properly set the low rank across different datasets? Is a rank of 20/30 a safe value for various datasets?

5. Is there any scenario where the proposed defense method fails? It would be great if the authors could find any case with deeper analysis.

---

### Official Review · Reviewer_9SoA · 2024-11-03

**Soundness:** 2
**Presentation:** 3
**Contribution:** 2
**Rating:** 5
**Confidence:** 5

**Summary:**

This paper presents Confidence Adaptation and Low-Rank compression (CALoR), a new defense strategy against model inversion attacks (MIAs). To mitigate MIAs, CALoR employs three key techniques: (1) confidence adaptation to reduce the confidence of private training samples, (2) low-rank compression to prevent attackers' ability from mitigating the overfitting, and (3) Tanh activation to induce gradient vanishing. Extensive experiments on both low-resolution and high-resolution attacks demonstrate the effectiveness of CALoR in defending against State-of-the-art MI attacks.

**Strengths:**

* The paper is well-written and easy to follow.

* Extensive experiments on both low-resolution and high-resolution attacks demonstrate the effectiveness of CALoR in defending against State-of-the-art MI attacks.

* Perform MI attacks  using modern model architectures like ViT-B/16 and Swin-v2.

**Weaknesses:**

* The experimental setup raises some questions. While PPA/MIRROR is primarily designed for high-resolution images, the experiments using PPA/ MIRROR are performed on low-resolution images. Given that StyleGAN-ADA generates 256x256 images, it's unclear how the authors adapt it to low resolution images (64x64). Similarly, PLGMI, originally designed for low-resolution attacks, is adapted for high-resolution scenarios. More details on this adaptation process should be explained.

* Section 3.2 demonstrates that reducing confidence in private training samples lowers attack accuracy. However, without reporting the model accuracy, it's difficult to determine if this reduction is solely due to a decrease in model utility. As existing work [r1] suggests a strong correlation between model accuracy and MI attack success, it's possible that any action diminishing model performance, including confidence reduction, could impact attack accuracy.

* The idea of reducing confidence in private training samples is similar to using label smoothing which is addressed in LS defense [r2]. Note that LS[r2] shows that positive label smoothing does not have such strong effect on defending against MI attacks.

[r1] Zhang, Yuheng, et al. "The secret revealer: Generative model-inversion attacks against deep neural networks." Proceedings of the IEEE/CVF conference on computer vision and pattern recognition. 2020.
[r2] Lukas Struppek,  et al. "Plug & play attacks: Towards robust and flexible model inversion attacks". In ICML, 2022.

**Questions:**

* To clarify the novelty of the proposed method, the authors should elaborate on the key differences between their approach and positive label smoothing [r2]. Without it, the contribution of the paper is limited.

* Could the authors provide a more comprehensive comparison with other state-of-the-art defenses, particularly high-resolution techniques like PPA and MIRROR, on the official benchmark?

* Could the authors include model accuracy metrics in Section 3.2 to better understand the trade-off between defense effectiveness and model utility?

* How does the proposed method perform on larger datasets like VGGFace/VGGFace2/CASIA?

---

### Official Review · Reviewer_KErs · 2024-11-04

**Soundness:** 3
**Presentation:** 3
**Contribution:** 2
**Rating:** 5
**Confidence:** 5

**Summary:**

This paper introduce a novel MI defence, namely CALoR, that aims to improve MI robustness by revisiting three potential weakness aspects of MI attack: MI objective, MI overfitting, and MI optimisation.

The empirical results are encouraging through various MI setups.

**Strengths:**

The presentation is clear and well organised.

The empirical results are encouraging.

**Weaknesses:**

1. While I appreciate the broad exploration of MI robustness in this paper, it feels more like a technical report rather than an in-depth investigation for each aspect. The proposed method consists of three modules: confidence adaptation, low-rank compression via a VAE-inspired framework, and Tanh activation. The concept of confidence modification has been somewhat investigated in LS defence, and the impact of vanishing gradients on MI is well-studied in PPA, LOMMA, and PLG-MI. These modules offer trivial and straightforward ways to mitigate MI attacks but lack novel insights or a significant contribution to MI.

2. The concept of confidence modification is somewhat similar to LS defence. If I understand it correctly, modifying the confidence by (positive) LS also offers the similar concept. However, I am quite surprise that (positive) LS and CA has opposite effect to MI. Could the authors provide an explanation on this?

3. While the experimental results are encouraging, the setups differ notably from those in existing studies. I suggest that the authors add additional experiments to strengthen the paper:
- For the low-resolution scenario, include LOMMA (rather than IF) as it is a state-of-the-art MI attack alongside PLG.
- For the high-resolution scenario, include PPA (rather than PLG) as it is a state-of-the-art MI attack alongside IF.
- The evaluation model used in the high-resolution scenario also differs from existing works. I am curious if this may affect the experimental setups.

4. For the ablation study, I recommend that the authors expand Table 3 to better demonstrate the effectiveness of each module in CALoR. The current results highlight the contribution of CA, but the roles of low-rank compression and Tanh activation remain unclear to me. To clarify, I suggest splitting the results for LoR into LoR with Tanh and LoR without Tanh.

5. The paper should include a paragraph addressing the weaknesses of the proposed method. One potential weakness is its complexity relative to existing defenses like LS or TL. With three different modules, the proposed approach may be challenging to adapt to new setups. As shown in the paper, in high-resolution scenarios with ImageNet as the pre-training dataset, CALoR demonstrates more difficulty in adapting compared to LS or TL.

**Questions:**

Please see the weakness

---

### Official Review · Reviewer_Ku5R · 2024-11-04

**Soundness:** 2
**Presentation:** 2
**Contribution:** 2
**Rating:** 5
**Confidence:** 4

**Summary:**

This work proposes a method for improving defense against generative Model Inversion Attacks (MIA), called Confidence Adaptation and Low-Rank Compression (CALoR). The proposed method demonstrates noticeable improvements over existing defense methods.

**Strengths:**

1) This paper is written well and it is easy to follow.

2) The proposed method obtains noticeable improvements (Table 1, Table 2) under IF and PLG-MI attacks.

**Weaknesses:**

1) Given that MIAs search for high-likelihood samples for a particular class, use of $L_{CA}$ basically manipulates the region where such high-likelihood samples are present, e.g..: Now it’s present in a region where $p\_c \approx 0.8$ and the MIA is searching in the region corresponding to $p_c~=1$. What would happen if the MIAs were aware of this confidence training? I suspect that a substantial amount of performance could be recovered if the identity loss is adjusted to align with $L_{CA}$ during model inversion. Conducting this ablation study is crucial.

2) **Using encoder features directly for MIA:** My understanding is that the proposed method focuses on white-box MIA attacks. What would happen if the adversary directly used the encoder features to perform MIA (by applying softmax directly to the encoder features) instead of the LORA features?

3) **User studies are necessary to show the efficacy of MI defense.** Since this work focuses on defending against private data reconstruction, it is important to conduct user study to understand the improvements (See [A, B]).

4) Why is the gap between TL and CALoR much smaller in the PLG/ImageNet setting compared to other setups (see Table 2)?

5) Error bars/ Standard deviation for experiments are missing.

6) It would be useful to indicate that the paper's focus is white-box MIAs.

7) Missing related works [A]

Overall I enjoyed reading this paper. But in my opinion, the weaknesses of this paper outweigh the strengths. But I’m willing to change my opinion based on the rebuttal.

=========

[A] Nguyen, Bao-Ngoc, et al. "Label-only model inversion attacks via knowledge transfer." Advances in Neural Information Processing Systems 36 (2024).

[B] [MIRROR] An, Shengwei et al. MIRROR: Model Inversion for Deep Learning Network with High Fidelity. Proceedings of the 29th Network and Distributed System Security Symposium.

**Questions:**

Please see Weaknesses section above for a list of all questions.

---

### Note · Authors · 2024-11-14

I have read and agree with the venue's withdrawal policy on behalf of myself and my co-authors.